# *GLW7.1*, a Strong Functional Allele of *Ghd7*, Enhances Grain Size in Rice

**DOI:** 10.3390/ijms23158715

**Published:** 2022-08-05

**Authors:** Rongjia Liu, Qinfei Feng, Pingbo Li, Guangming Lou, Guowei Chen, Haichao Jiang, Guanjun Gao, Qinglu Zhang, Jinghua Xiao, Xianghua Li, Lizhong Xiong, Yuqing He

**Affiliations:** National Key Laboratory of Crop Genetic Improvement and National Centre of Plant Gene Research, Hubei Hongshan Laboratory, Huazhong Agricultural University, Wuhan 430070, China

**Keywords:** *GLW7.1*, GHD7, grain size, quality, rice

## Abstract

Grain size is a key determinant of both grain weight and grain quality. Here, we report the map-based cloning of a novel quantitative trait locus (QTL), *GLW7.1* (*Grain Length*, *Width and Weight 7.1*), which encodes the CCT motif family protein, GHD7. The QTL is located in a 53 kb deletion fragment in the cultivar Jin23B, compared with the cultivar CR071. Scanning electron microscopy analysis and expression analysis revealed that *GLW7.1* promotes the transcription of several cell division and expansion genes, further resulting in a larger cell size and increased cell number, and finally enhancing the grain size as well as grain weight. *GLW7.1* could also increase endogenous GA content by up-regulating the expression of GA biosynthesis genes. Yeast two-hybrid assays and split firefly luciferase complementation assays revealed the interactions of GHD7 with seven grain-size-related proteins and the rice DELLA protein SLR1. Haplotype analysis and transcription activation assay revealed the effect of six amino acid substitutions on GHD7 activation activity. Additionally, the NIL with *GLW7.1* showed reduced chalkiness and improved cooking and eating quality. These findings provide a new insight into the role of *Ghd7* and confirm the great potential of the *GLW7.1* allele in simultaneously improving grain yield and quality.

## 1. Introduction

Rice (*Oryza sativa* L.) is the most important staple food crop in the world and feeds more than half of the world’s population [1]. Therefore, to meet the food needs of a rapidly growing global population, increasing rice grain yield has been a major breeding goal. Rice yield is mainly determined by three major components: grain weight, number of grains per panicle and number of effective tillers per plant [2]. Among them, grain weight is largely determined by grain size, which includes grain length, width, and thickness [3]. In recent decades, many quantitative trait loci (QTLs)/genes regulating grain size have been isolated and shown to participate in multiple signaling pathways, including the G-protein signaling pathway, the ubiquitin–proteasome pathway, mitogen-activated protein kinase (MAPK) signaling pathway, phytohormone signaling and homeostasis, and transcriptional regulators [4].

G proteins are guanine nucleotide-binding trimeric proteins consisting of Gα, Gβ and Gγ subunits and regulate many biological processes. *GRAIN SIZE 3* (*GS3*), encoding an atypical Gγ protein, is the first identified major QTL negatively regulating grain size [5]. DENSE AND ERECT PANICLE 1 (DEP1), another atypical Gγ protein, positively regulates grain size by competitively binding to Gβ (RGB1) with GS3 [6]. Other G-proteins, including conventional Gγ proteins (RGG1 and RGG2), atypical Gγ protein (GGC2), Gα protein (RGA1) and Gβ protein (RGB1) could also control grain size [6,7,8]. In addition, OsMADS1, a MADS-domain transcription factor, negatively regulates grain length through directly interacting with GS3 and DEP1 [9].

Ubiquitination and deubiquitination are two opposite protein modifications which are involved in the regulation of rice grain size. GRAIN WIDTH 2 (GW2), a RING-type E3 ubiquitin ligase, negatively regulates grain width by ubiquitinating WG1 and targeting it for degradation via the 26S proteasome pathway [10]. CHANG LI GENG 1 (CLG1), another RING-type E3 ubiquitin ligase, positively regulates grain length by ubiquitinating GS3 and targeting it for degradation via the endosome pathway [11]. *WIDE AND THICK GRAIN 1* (*WTG1*), which encodes an otubain-like protease with deubiquitination activity, controls grain size and shape mainly by affecting cell expansion in the spikelet hull [12]. *LARGE GRAIN 1* (*LG1*), which encodes a constitutively expressed ubiquitin-specific protease15 (OsUBP15) with deubiquitination activity, positively regulates grain width and size [13].

The OsMKKK10-OsMKK4-OsMPK6 cascade has been revealed to positively regulate grain size by promoting cell proliferation in spikelet hulls [14,15,16]. GRAIN SIZE AND NUMBER 1 (GSN1)/OsMKP1, a MAPK phosphatase, negatively regulates grain size by directly interacting with and inhibiting the dephosphorylation of OsMPK6 [17]. In addition, the upstream gene *OsER1*, which encodes a receptor-like protein kinase, and the downstream transcription factor *OsWRKY53*, could both positively regulate grain size through the MAPK signaling cascade [18,19].

Phytohormones play various roles in plant growth and development, stress responses, and metabolism. Several genes controlling grain size have been reported to be involved in the brassinosteroid (BR) signaling pathway, such as *GRAIN WIDTH 5* (*GW5*) [20], *GRAIN LENGTH 2* (*GL2*) [21,22] and *GRAIN LENGTH 3.1* (*GL3.1*) [23]. Another set of genes regulate grain size through the auxin signaling pathway, such as *THOUSAND GRAIN WEIGHT 6* (*TGW6*) [24], *BIG GRAIN 1* (*BG1*) [25] and *THOUSAND GRAIN WEIGHT 3* (*TGW3*) [26]. Moreover, some gibberellic acid (GA)-signaling-pathway-related genes also regulate grain size, such as *GIBBERELLIN-DEFICIENT DWARF 1* (*GDD1*) [27], *SMALL AND ROUND SEED 3* (*SRS3*) [28] and *SMALL GRAIN AND DWARF 2* (*SGD2*) [29].

Many transcription factors participate in the regulation of grain size, including the SQUAMOSA promoter binding protein-like (SPL) family (*GRAIN LENGTH AND WIEIGHT 7* (*GLW7*)/*OsSPL13*, *GRAIN WIDTH 8* (*GW8*)/*OsSPL16*, *OsSPL18*) [30,31,32], the basic helix–loop–helix (bHLH) family (*Awn-1* (*An-1*), *OsbHLH079, OsbHLH107*) [33,34,35], APETALA2-type (AP2) transcription factors (*SMALL ORGAN SIZE1* (*SMOS1*), *SUPERNUMERARY BRACT* (*SNB*), *FRIZZY*
*PANICLE* (*FZP*)) [36,37,38], and other transcription factors (*GRAIN SHAPE 9* (*GS9*), *SHORT GRAIN6* (*SG6*), *GRAIN LENGTH 4* (*GL4*)) [39,40,41].

Although many QTLs/genes regulating grain size have been identified, the understanding of grain size regulation is still fragmented. In this study, we report the mapping, cloning and initial characterization of a novel grain size QTL, *GLW7.1* (*Grain Length, Width and Weight 7.1*) in rice, which encodes the CCT (CONSTANS, CONSTANS-LIKE, and TIMING OF CHLOROPHYLL A/B BINDING1) motif family protein, GHD7. *Grain number, plant height, and heading date 7* (*Ghd7*) was first reported as a major regulator of heading date, and improved yield by increasing grain number [42]. Subsequent studies revealed that it participated in a variety of other developmental processes, such as stress responses, seed germination and nitrogen utilization [43,44,45]. Here, we performed scanning electron microscopic analysis, yeast two-hybrid assays, split firefly luciferase complementation (SFLC) assays and expression analysis to uncover the mechanism mediated by *GLW7.1* to regulate grain size. We also conducted haplotype analysis of *Ghd7* and transcription activation assay to uncover the reason underlying different effects between three allelic GHD7 proteins. Our results provide insights into the role of *Ghd7* in regulating grain size and the effect of different amino acid substitutions on transcriptional activation activity of GHD7 proteins, and we provide a promising *Ghd7* allele for breeding rice with high yield and superior quality.

## 2. Results

### 2.1. Identification of GLW7.1

To identify novel QTLs associated with grain size (Figure 1A), we selected two *indica* varieties, Jin23B (hereafter J23B) and CR071, that showed significant differences in grain size (Figure 1B,C) and constructed a set of 238 BC_3_F_1_ lines in the J23B background (Appendix A). Three QTLs were revealed by a subsequent QTL analysis, among which the QTL located between SSR markers RM501 and RM542 on chromosome 7 was the most significant contributor to grain length (Figure 1A, Appendix A). To further evaluate the genetic effect of this QTL, we developed a near-isogenic line (NIL) in the genetic background of J23B (Appendix A). Genetic analysis of BC_4_F_2_ progenies derived from the NIL in 2015 showed that the dominant allele from CR071 could increase grain length, grain width and grain weight (Appendix A). The similar genetic effect was further confirmed by BC_5_F_3_ progenies in 2017 (Appendix A). Thus, we designated this QTL as *Grain Length*, *Width and Weight 7.1* (*GLW7.1*).

### 2.2. Characterization of GLW7.1

Two NIL plants carrying the homozygous J23B allele *glw7.1* and CR071 allele *GLW7.1* were developed and named as NIL-J and NIL-C, respectively. Compared to NIL-J, NIL-C displayed a higher value in grain length (increased by 8%) (Figure 1B,E) and grain width (increased by 5%) (Figure 1C,F), leading to an increase in length-to-width ratio by 2% (Figure 1G) and 1000-grain weight by 22% (Figure 1H). The plant height of NIL-C was about 33 cm higher than that of NIL-J (Figure 1D,I), but no difference was observed in tiller numbers per plant (Figure 1J). Meanwhile, NIL-C displayed more filled grains per panicle (increased by 80%) than NIL-J (Figure 1K). Thus, the increase in grain weight and grain number contributed to the increase in grain yield per plant by 114% in NIL-C, in comparison with NIL-J (Figure 1L).

The dominant *GLW7.1* locus with a yield-increasing effect has a good advantage in hybrid rice breeding. Considering that the simultaneous increase in rice grain length and width is usually accompanied by a decrease in rice quality [46,47], to further evaluate the prospects of *GLW7.1* in rice breeding, we then examined rice quality traits among NILs, including percentage of grains with chalkiness, amylose content, gel consistency, and taste value. Surprisingly, a significant reduction in grain chalkiness and a huge improvement in taste score were observed in NIL-C plants (Figure 2A,B,E), accompanied with a significant increase in amylose content and gel consistency (Figure 2C,D). These results demonstrate that the *GLW7.1* allele from CR071 is a pleiotropic gene conferring high yield and superior quality.

The glume, including lemma and palea, determines the upper limit of grain size [39,46,48,49], and its size is determined by cell number and cell size. To uncover the cytological reason underlying the difference in grain size between NIL-J and NIL-C, we performed scanning electron microscopic analysis of the outer surfaces of lemmas (Figure 3A,B). Compared with NIL-J, the value of cell length, cell width and the number of longitudinal cells were significantly higher in NIL-C (Figure 3C–E), but the number of transverse cells showed no difference (Figure 3F). To further investigate how *GLW7.1* regulates cell number and cell size, we examined the expression levels of 43 genes involved in cell cycle and cell expansion using the young panicles (8–10 cm in length) of the two NILs. As expected, expression levels of 10 cell cycle related-genes (*CYCD1;1*, *E2F*, *MCM4*, *CDC20*, *CYCA2;3*, *CYCB1;1*, *CYClaZm*, *MAPK*, *CDKB* and *KN*) and 3 cell-expansion-related genes (*EXPA3*, *EXPA5* and *EXPB3*) were significantly up-regulated (fold-change > 1.5 and *p* < 0.01) in NIL-C (Figure 3G, Appendix A). These results suggest that *GLW7.1* positively regulates grain size by promoting cell division and cell expansion to increase cell number and cell size of the glume during spikelet development.

### 2.3. Fine Mapping of GLW7.1

To fine-map *GLW7.1*, we developed a random population consisting of 30,000 individuals from NIL-H lines (NIL plants with heterozygous allele *GLW7.1/glw7.1*) and screened recombinants in the target region using two newly developed markers (G7.1 and LG15). A total of 600 recombinants were identified, and further genotyping was conducted using 16 newly developed simple sequence repeats (SSR) and kompetitive allele-specific PCR (KASP) markers (Figure 4A, Appendix A). The grain size of the 600 recombinants and 70 non-recombinants derived from the random population was investigated, and *GLW7.1* was mapped to the interval between LG18 and K5 by a subsequent QTL analysis (Appendix A). Subsequently, we performed a progeny test by investigating the grain size of homozygous progenies derived from each recombinant, and three non-recombinant lines (NIL-J, NIL-C and NIL-H) were designated as controls. In the end, the *GLW7.1* locus was narrowed to the region between markers K17 and K19 (Figure 4B and Appendix A).

By comparing the genomic sequences of Nipponbare (http://rice.uga.edu/, accessed on 18 March 2019), Zhenshan97 and Minghui63 (https://rice.hzau.edu.cn/rice_rs3/, accessed on 18 March 2019), we found a large fragment (~38 kb/55 kb) insertion between Zhenshan97 and Nipponbare/Minghui63 in the candidate region between markers K17 and K19 (Appendix A). In order to fine-map the candidate gene, the whole genome of J23B and CR071 were separately sequenced on Illumina and Nanopore (ONT) platforms to capture the target candidate segment sequences. Compared with CR071, J23B contained a 53 kb deletion in the candidate segment between markers K17 and K19 (68 kb in J23B and 121 kb in the CR071) (Appendix A).

Three predicted open reading frames (ORFs) (*ORF1*, *ORF2* and *ORF4*) were located in the 68 kb target region of J23B, and four predicted ORFs (*ORF1*, *ORF2*, *ORF3* and *ORF4*) were located in the corresponding 121 kb region of CR071, excluding those ORFs encoding transposon and retrotransposon proteins (Figure 5A). *ORF1*, *LOC_Os07g15670*, encodes a putative peroxiredoxin. *ORF2*, *LOC_Os07g15680*, encodes a putative phospholipase D. *ORF3*, *LOC_Os07g15770*, encodes a CCT motif family protein, GHD7, which was reported to regulate heading date and yield potential in rice [42] (Figure 5B). *ORF4*, *LOC_Os07g15820*, encodes an expressed protein with unknown function.

### 2.4. Positional Cloning of GLW7.1

To ascertain the candidate gene underlying *GLW7.1*, we compared the genomic sequences of the three ORFs (*ORF1*, *ORF2* and *ORF4*) from J23B and CR071, including promoter regions and protein-coding regions, and found no variation. Therefore, *ORF3* or *Ghd7*, which is located in the 53 kb deleted region of J23B, was likely to be the candidate of *GLW7.1*. We subsequently compared the genomic sequences of *Ghd7* from Minghui63, CR071 and Nipponbare. Different from the reported *Ghd7-1* allele of Minghui63, the allele of CR071 was termed as *Ghd7-3* because of three amino acid substitutions, and the allele of Nipponbare was termed as *Ghd7-2* because of four amino acid substitutions. The allele of J23B and Zhenshan97 was termed as *Ghd7-0* because of the loss of the complete gene region [42] (Figure 5C).

To determine whether the *Ghd7-3* allele underlies the QTL *GLW7.1*, we conducted a knockout experiment by editing the allele using the CRISPR-Cas9 system in the background of NIL-C. The sequence (c. 512 TGGCCAATGTTGGGGAGAGC) in the second exon was designed as the sgRNA target site to produce mutations neighboring the CCT domain coding region (Figure 6A). Three mutated alleles were obtained, named A1, A4 and A8. The allele A4 showed minor amino acids change (AN → D) and still retained the CCT domain. By contrast, both 1 bp insertion in allele A1 and 20 bp deletion in allele A8 resulted in frameshift mutations, which caused the loss of the CCT domain (Figure 6B). As expected, the alleles A1 and A8 produced smaller grains than the allele A4 in the NIL-C background (Figure 6C–F). We also conducted a complementation experiment by expressing the cDNA of the *Ghd7-3* allele driven by its native promoter in the NIL-J background. The grains produced from the complemented lines Com1, Com2 and Com3 were larger than those from negative transgenic plants NIL-J-Neg (Figure 6C–F). Additionally, the other two alleles of *Ghd7* (*Ghd7-1* and *Ghd7-2*) could also increase grain size in the Zhenshan97 background (Appendix A). Together, these data indicate that *Ghd7-3* is the functional gene underlying *GLW7.1*.

### 2.5. Haplotype Analysis of Ghd7

In order to investigate natural variations in *Ghd7*, we analyzed the sequencing data of 533 core germplasms in the *Ghd7* region [50]. Based on the nonsynonymous polymorphisms in the coding region that lead to amino acid substitutions or protein premature truncation (Appendix A), nine haplotypes of *Ghd7* can be identified (Figure 7A), in agreement with the types reported [51]. Of these, four major haplotypes contained 498 accessions, while the remaining five rare haplotypes contained only 27 accessions. The four major haplotypes of *Ghd7* have been reported to be strong function, weak function and loss-of-function, respectively [42]. Three major haplotypes mainly existed in *indica*: Hap1 (*Ghd7-1*) represented by Minghui63 and 9311 was the type with strong function, Hap2 (*Ghd7-3*) represented by Teqing, and CR071 was another type with strong function, and the Hap9 (*Ghd7-0*) represented by Zhenshan97 and J23B was the type with loss of function. In addition, Hap4 (*Ghd7-2*) represented by Nipponbare and Zhonghua11, was the major haplotype with weak function in *japonica*.

Given that the middle region of CCT domain proteins was previously reported to have transactivation activity [52], we performed transcription activation assay in rice protoplasts prepared from leaf sheath of Nipponbare seedlings to verify the GHD7-mediated activation among three haplotypes. The three allelic GHD7 proteins (Hap1, Hap2 and Hap4 were respectively derived from Minghui63, CR071 and Nanyangzhan) were fused to the GAL4 DNA-binding domain (GAL4DBD) to generate effectors, and the firefly luciferase gene (*LUC*) was used as the reporter (Figure 7B) [53]. Compared with the GAL4 negative control, the three allelic GHD7 proteins showed dramatically different activation activity (Figure 7B and Appendix A). The protein GHD7-Hap4 exhibited the strongest activation activity, compared with the weaker activity exhibited by GHD7-hap1 and the weakest activity by GHD7-Hap2. The difference in activation activity of the three allelic proteins may be due to amino acid substitutions.

To test this hypothesis, we added another four allelic GHD7 proteins (Hap3, artificial HapN1 combining exon1 region of Hap1 with exon2 region of Hap2, artificial HapN2 combining exon1 region of Hap1 with exon2 region of Hap4, and artificial HapN3 combining exon1 region of Hap2 with exon2 region of Hap4) to generate effectors (Figure 7B). A comparison of GHD7-Hap1 and GHD7-Hap3 revealed that the A233P substitution seems to have no effect on activation activity (Figure 7B and Appendix A), consistent with its position in the CCT DNA-binding domain. By comparing GHD7-Hap2 and GHD7-HapN1, GHD7-HapN2 and GHD7-HapN3, we found that the A111G substitution seems to also have no effect on activation activity (Figure 7B and Appendix A). A comparison of GHD7-Hap3, GHD7-Hap4 and GHD7-HapN2 revealed that both the G122E/S136G and V174D substitutions could strengthen the activation activity (Figure 7B and Appendix A). By contrast, the D173N substitution could weaken the activation activity, by comparing GHD7-Hap3 and GHD7-HapN1 (Figure 7B and Appendix A). These results suggest that the different genetic effects of the three major haplotypes may be due to the transcription activation activity difference among GHD7 proteins, which was promoted when G122E/S136G and V174D substitutions were contained and weakened when D173N substitution were contained.

### 2.6. GLW7.1 Determines Grain Size via Grain-Size Genes

To uncover the molecular pathway by which *GLW7.1* regulates grain size, we conducted a yeast two-hybrid (Y2H) screen using the C-terminal of GHD7 (aa. 208–257) as bait and a normalized prey library derived from young panicles of Zhenshan97. A total of 325 candidate positive clones were detected on synthetic growth medium without leucine, tryptophan, histidine and adenine. Of them, seven grain-size proteins (OsFBK12, FZP, OsNAC024, OsNAC025, OsNF-YC12, RICE STARCH REGULATOR 1 (RSR1) and SNB) [37,38,54,55,56,57] and the rice DELLA protein SLENDER RICE 1 (SLR1) [58] were selected for examination by reconstructing the prey vectors with the full-length protein sequences. The interactions of GHD7 with these proteins were then confirmed by X-α-gal filter lift assays (Figure 8A). Furthermore, the interactions were also demonstrated using split firefly luciferase complementation (SFLC) assays in tobacco leaf epidermal cells (Figure 8B). These results imply that the GHD7 protein may regulate the grain size and weight through interactions with the above grain-size proteins and DELLA protein.

To explore the downstream genes of *GLW7.1* in regulating grain size, we detected the expression levels of 63 grain-size genes in NIL-J and NIL-C by qRT-PCR analysis using the young panicles (8–10 cm in length). The *GLW7.1* locus significantly up-regulated the expression of nine positive grain-size genes (fold-change > 1.5 and *p* < 0.01), including genes encoding OsBZR1 (BES1/BZR1 homolog protein) [59], OsMAPK6 [15], GLW7 [31], OsbHLH107 [35], IDEAL PLANT ARCHITECTURE 1 (IPA1) [60], SRS3 (a kinesin motor domain protein) [61], SMALL AND ROUND SEED 5 (SRS5) (alpha-tubulin protein) [62] and OsWRKY53 [19] (Figure 9A). Furthermore, we observed an extremely significant down-regulation of *OsMADS1* (fold-change = 56.7 and *p* = 6.1 × 10^−7^), which negatively regulates grain length [9] (Figure 9A). Furthermore, transcription activation assay in rice protoplasts prepared from leaf sheath of Zhenshan97 seedlings shown that the LUC activity driven by the *OsMAPK6* promoter was significantly induced by GHD7 (Figure 9B), which indicates that GHD7 could directly activate the expression of *OsMAPK6*. Overall, these results suggest that *GLW7.1* positively regulates grain size through a series of grain-size genes.

### 2.7. GLW7.1 Positively Regulates GA Biosynthesis

Mutants with defection in GA biosynthesis, such as *gdd1* and *sgd2*, usually show reduced height and small seeds [27,29]. Using quantitative RT-PCR, we detected the up-regulated expression of four GA biosynthetic genes (*KS1*, *KO2*, *KAO* and *GA3ox2*), GA catabolic gene (*GA2ox3*) and the GA signaling pathway gene *SLR1* in the young panicles of NIL-C, compared to NIL-J (Figure 10A). To examine whether *Ghd7* was involved in GA biosynthesis, we analyzed the response of NIL-J and NIL-C to exogenous GA_3_ and paclobutrazol (PAC, a GA biosynthesis inhibitor) treatment. The length of the second leaf sheath of NIL-J was significantly shorter than NIL-C and could be restored to the NIL-C level by exogenous GA_3_ treatment (Figure 10B,C). In addition, the growth of both NIL lines was simultaneously inhibited by exogenous PAC treatment, and their second leaf sheaths were almost the same in length (1.51 cm in NIL-J and 1.41 cm in NIL-C) (Figure 10B,C). Moreover, the second leaf sheath of NIL-J was almost as long as NIL-C with different GA_3_ concentrations treatment (Appendix A). Collectively, these findings suggest that *Ghd7* may be involved in GA biosynthesis rather than GA response.

To further confirm the role of *Ghd7* in GA biosynthesis, we measured endogenous GA_1_ levels in 2-week-old seedlings of NIL-J and NIL-C. The GA_1_ level in NIL-J was approximately 84.6% (0.69 ng/g) of that in NIL-C (0.81 ng/g) (Figure 10D). The effect of exogenous GA_3_ treatment on *Ghd7* expression was also investigated by quantitative RT-PCR. The diurnal expression pattern of *Ghd7* was consistent with the previous report [42], and the expression level of *Ghd7* was significantly inhibited at 6 h after the exogenous GA_3_ treatment (Figure 10E). Interestingly, the expression levels of two GA biosynthetic genes, *GA20ox2* and *GA3ox2*, were also reported to be inhibited after the exogenous GA3 treatment [27]. Together, these results suggest that *Ghd7* positively regulates endogenous GA biosynthesis.

## 3. Discussion

### 3.1. GLW7.1 Simultaneously Improves Grain Yield and Quality

In this study, we performed the map-based cloning of *GLW7.1*, a novel QTL regulating grain size, and confirmed that *Ghd7*, encoding a CCT motif family protein, underlies the QTL. *Ghd7* was previously reported as a major regulator of heading date and improved yield by increasing grain number [42]. Our results demonstrate that *GLW7.1,* or *Ghd7-3,* not only increases grain number, but also increases grain weight, manifested as increases in both grain length and width (Figure 1). On the other hand, *GLW7.1* has effects on reducing chalkiness and improving cooking and eating quality (Figure 2). Thus, *GLW7.1* or *Ghd7-3* is a positive regulator of both rice yield and quality. In contrast, many genes regulating grain size have negative effects on rice quality. For example, *GS2* and *GW2* increase not only grain size and weight, but also chalkiness simultaneously [46,47]. In addition, only 73 out of 533 accessions carry the *GLW7.1* allele (Figure 7A). Therefore, *GLW7.1* is a promising allele for simultaneously improving grain yield and quality during rice breeding.

### 3.2. Natural Variations Alter the Transcriptional Activity of GHD7

In this study, we performed haplotype analysis of *Ghd7* using a germplasm population consisting of 533 accessions and identified nine haplotypes based on nonsynonymous SNPs in the coding region (Figure 7A). Among those, the three reported functional alleles with different effects on heading date were included, which were Hap1 (*Ghd7-1*), Hap2 (*Ghd7-3*) and Hap4 (*Ghd7-2*) [42]. In order to uncover the reason underlying different effects of the three allelic GHD7 proteins, we focused on the six amino acid substitutions among them. The subsequent transcription activation activity assay showed that the G122E/S136G and V174D substitutions strengthen the activation activity, the D173N substitution weakens it, and the A111G and A233P substitutions have no effect on it (Figure 7B and Appendix A). It has been demonstrated that acidic amino acids in the activation domain are essential for the transcriptional activation of transcription factors. For example, when all the acidic amino acid residues in the activation domain of transcription factor OCT4 were replaced by alanine, its transcriptional activation activity decreased dramatically [63]. In our study, Hap4 allelic GHD7 contains two amino acid substitutions that increase acidic amino acids, G122E and V174D; Hap2 allelic GHD7 contains an amino acid substitution that decreases the acidic amino acid, D173N. Coincidentally, these acidic amino acid substitutions in the activation domain also cause similar changes in GHD7 transcriptional activation.

Furthermore, we noticed the transcriptional repression activity of GHD7 on *ARE1* [44], which seems to contradict our results. The study investigated the transcriptional repression of GHD7 on a specific downstream gene, *ARE1*, but did not investigate the transcriptional activation or repression activity of GHD7 itself. Actually, there was one study investigating the transcriptional repression activity of GHD7 [45]. Weng et al. found that the transcriptional activation activity of GAL4-VP16-GHD7 fusion protein was significantly weaker than that of GAL4-VP16, and they draw the conclusion that GHD7 had intrinsic transcriptional repression activity. We thought the conclusion was not rigorous because GHD7 fused to the C-terminal of VP16 may weaken the transcriptional activation activity dependent on VP16 C-terminal, and the transcriptional repression activity may not be caused by GHD7, but by C-terminal fusion. In our study, we investigated the transcriptional activation activity of GHD7 itself (Figure 7B and Appendix A) and observed its transcriptional activation of *OsMAPK6* (Figure 9B). However, it was reported that the ABI4 protein has different transcriptional activity for different downstream target genes. ABI4 could bind to promoters of *GA2ox7* and *NCED6* to activate the expression of these two genes [64], and could also bind to promoters of *CYP707A1* and *CYP707A2* to inhibit the expression of these two genes [65]. We hypothesized that GHD7 may function in a similar way on target genes.

In conclusion, natural variations in GHD7 proteins affect its transcriptional activity, which is likely to influence transcription of the downstream genes, and finally result in different effects on heading date and other traits. In addition, there are another three amino acid substitutions with unknown effects on its activation activity (Figure 7A), which may endue the remaining four haplotypes with different functions. Therefore, it would be of great importance to reveal the effect of each amino acid substitution and further construct different combinations of these natural variations, which may provide new ideas for future applications of *Ghd7* in rice breeding. 

### 3.3. The Pathway of GLW7.1 Controlling Grain Size

In this study, for the first time, we showed that *Ghd7* is a major gene regulating grain size, which has long been recognized as a major regulator of heading date in rice [42,51]. Recently, the interactions of GHD7 with the CCAAT-box-binding transcription factors, OsNF-YB11 and OsNF-YC2, have been elucidated [66]. Similar to the interactions between GHD7 and OsNF-YB11/OsNF-YC2, a physical interaction between GHD7 with OsNF-YC12, another CCAAT-box-binding transcription factor, was detected in yeast cells and tobacco leaf epidermal cells (Figure 8). For *OsNF-YC1**2*, the mutants showed a decrease in grain length and width, and the overexpression showed an increase, implying a positive regulation in grain size [56]. We also found the interactions between GHD7 with three AP2 domain containing proteins, FZP, RSR1 and SNB (Figure 8) [37,38,57]. Of the three, FZP positively and SNB negatively regulate grain length and width by simultaneously affecting cell proliferation and expansion in spikelet hulls. Additionally, OsFBK12, an F-box protein containing a Kelch repeat motif, which positively regulates seed size by increasing cell size but decreasing cell number [54], and two NAC-type TFs, OsNAC024 and OsNAC025, which bind to the promoters of three grain size/weight regulating genes (*GW2*, *GW5* and *DWARF 11* (*D11*)) to modulate grain size [55], were also observed in the yeast two-hybrid assays and SFLC assays (Figure 8).

Expression analysis revealed that *GLW7.1* up-regulated the transcription of eight genes having positive effects on grain size, and down-regulated that of *OsMADS1*, a negative regulator of grain size, in NIL-C panicles (Figure 9A). Interesting, the expression of *OsMADS1* was significantly reduced in the mutants of *SNB* and *FZP*, which encode proteins interacting with GHD7, implying *OsMADS1* may be downstream of these two genes [67,68]. In addition, cytological observations showed that *GLW7.1* enhances grain size by promoting cell proliferation and expansion (Figure 3A–F). Among the up-regulated positive regulation genes of grain size, *OsBZR1* can affect grain length and grain width simultaneously, which are located at the downstream of BR signaling pathway [59]. The OsSPL13 protein encoded by *GLW7* could bind to the promoter of *SRS5* and activate its expression, and the two genes could promote glume cell expansion and jointly regulate grain length [31,62]. *OsMAPK6* and *OsWRKY53*, both of which belong to the MAPK signaling cascade, could promote cell proliferation and thus increase grain length and width in rice [15,19]. *OsMADS1*, a significantly down-regulated negative regulation gene of grain length, also regulates rice grain length by affecting cell proliferation [9]. However, the transcription activation activity assay showed that GHD7 could directly activate the expression of *OsMAPK6* (Figure 9B), which indicates that GHD7 may directly bind to the OsMAPK6 promoter to regulate its expression and thus control rice grain size.

We also investigated the expression of several genes that regulate cell cycle and cell expansion and found significantly higher expression levels of 10 cell-cycle-related genes and 3 cell expansion related-genes in NIL-C panicles (Figure 3G). However, we noted the up-regulation of three negative grain-size genes and the down-regulation of one expansion gene, which may indicate some unclear regulatory mechanisms. Therefore, based on the results above, we proposed that GHD7 interacts with several grain-size-related proteins, up-regulates the positive regulation genes of grain size, such as *OsMAPK6*, and down-regulates the negative, *OsMADS1*, thereby promoting the transcription of downstream cell division and expansion genes, and finally enhancing the grain size as well as grain weight (Appendix A).

### 3.4. GHD7 Participates in GA Pathway

In the GA signaling pathway, GA binds to the GID1 receptor, leading to the formation of a GID1-GA-DELLA complex, which further stimulates the interaction of DELLA with the SCF^GID2^ complex. Once recruited to SCF^GID2^ complex, DELLA is polyubiquitylated and then subsequently degraded through the 26S proteasome pathway [69]. Mutants with defection in the GA signaling pathway usually show reduced height and small seeds. For example, *GDD1* encodes a kinesin-like protein that directly regulates the expression of the *KO2* gene [27], and *SGD2* encodes an HD-ZIP II transcription factor that positively regulates the expression of GA biosynthesis genes [29], while their mutants reduce endogenous GA levels, leading to a decrease in cell size, thus resulting in a severely dwarfed, small-grain phenotype. Moreover, overexpressing RGG2, which mediates internal GA biosynthesis and participates in the GA signaling pathway, also causes dwarfism and small grains [7]. Moreover, the grain-size gene that was upregulated most in NIL-C, SRS3 (Figure 9A), the mutant of which showed reduced height and short seeds, was also reported to participate in regulating the expression of genes in the GA biosynthesis pathway [28]. In this study, we also observed that *Ghd7* could positively regulate endogenous GA biosynthesis (Figure 10).

In *Arabidopsis*, DELLA proteins physically interact with the CCT domain of CONSTANS (CO) and integrate gibberellic acid and photoperiod signaling to regulate flowering under long days [70]. Here, we identified the interactions of GHD7 with the rice DELLA protein, SLR1 (Figure 8), which has been reported to interact with transcription factors, such as GRF4-GIF1, NACs and OsPIL14, and inhibit their transcriptional activation of downstream genes [71,72,73]. Comprehensively considering the pleiotropic effect of *Ghd7* on plant height and grain size (Figure 1, Figure 6 and Appendix A), we hypothesized that GHD7 participates in the GA biosynthesis to increase grain size (Appendix A) and is regulated by the GID1-GA-DELLA module as feedback of the pathway.

## 4. Materials and Methods

### 4.1. Plant Materials and Growth Conditions

Two *indica* cultivars Jin23B (J23B) and CR071 were used to construct the QTL mapping population, with details described in Appendix A. The NIL populations used for the genetic analysis of *GLW7.1* in grain size in two years were, respectively, isolated from BC_4_F_1_ and BC_5_F_2_ plants with heterozygous allele *GLW7.1/glw7.1*. The progeny test was conducted in the BC_5_F_4_ generation. The NILs (NIL-J and NIL-C) involved in a series of subsequent experiments were isolated from the BC_5_F_n_ (*n* ≥ 6) plant with heterozygous allele *GLW7.1/glw7.1*. The NIL-J-Com lines were obtained by complementation test. The mutants (NIL-C-A lines) were obtained by CRISPR/Cas9-based genome editing. The T_1_ generation materials were used for the analysis. The *Ghd7-2* allele lines (NIL-NYZ and NIL-ZS) were kindly provided by Guangming Lou. The Ghd7-1 allele lines (ZS-Com and ZS-Neg) were kindly provided by Lei Wang. The rice plants were grown at the experimental field of Huazhong Agricultural University in Wuhan during the summer with a density of 16 cm × 26 cm under normal field management.

### 4.2. Trait Measurement

Fully filled grains from each plant were used for measuring grain length, grain width, grain number, grain yield and 1000-grain weight by the yield traits scorer (YTS) platform [74] after 2014. (The phenotype of grain length for primary QTL mapping was measured by an electronic digital display caliper in 2012.) The plant height was measured from the main culm. The number of tillers per plant was counted as all fertile panicles in one plant. The percentage of grains with chalkiness, amylose content and gel consistency were measured according to the NY/T 593-2013 standard published by the Ministry of Agriculture, China (http://www.zbgb.org/27/StandardDetail1476335.htm, accessed on 3 October 2019). The taste score of milled rice was evaluated using a taste analyzer kit (Satake, RLTA10B-KC, Hiroshima, Japan) [75].

### 4.3. Linkage Analysis and QTL Mapping

A primary mapping population of 238 BC_3_F_1_ individuals was generated from a cross between J23B and CR071 (Appendix A). The plants were then genotyped by 157 polymorphic SSR markers covering the whole genome. The grain length was measured and the linkage analysis was carried out by composite interval mapping module of the software WinQTLCart 2.5 [76]. The R/qtl package [77] was employed to plot the linkage map using the output of WinQTLCart.

For fine mapping of the *GLW7.1*, we developed a BC_5_F_3_ population consisting of 30,000 individuals from NIL plants with heterozygous allele *GLW7.1*/*glw7.1*. *GLW7.1* was mapped to the interval between LG18 and K5 by a subsequent linkage analysis, and then narrowed to the region between markers K17 and K19 by progeny test. Relevant primer sequences were listed in Appendix A.

### 4.4. Scanning Electron Microscopy

Lemmas of spikelets at the heading stage were collected and fixed in FAA solution (50% ethanol, 5% glacial acetic acid and 3.7% formaldehyde) for more than 16 h. The fixed samples were dehydrated in a graded ethanol series and then critical point dried, followed by being coated with gold. The samples were then observed with a scanning electron microscope (JEOL, JSM-6390LV, Tokyo, Japan) at an accelerating voltage of 10 kV and a spot size of 30 nm. The morphology of lemma cells was scanned at a magnification of 100× to measure cell length and cell width, and at 50× with three pictures that are combined to cover the entire lemma to measure the cell number. The cell size of the lemmas was measured from pictures using ImageJ software (NIH), and the cell number was counted manually.

### 4.5. RNA Extraction and Expression Analysis

Total RNA was extracted from young panicles (8–10 cm in length) and 2-week-old seedlings using TRIzol reagent (Invitrogen, 15596026, Shanghai, China). DNase I (Invitrogen, 18068015, Shanghai, China) pre-treated RNA was reverse-transcribed using the M-MLV Reverse Transcriptase kit (Promega, M170A, Madison, WI, USA) following the manufacturer’s instructions. The qRT–PCR was then conducted in a total volume of 10 μL, which consisted of 5 μL of cDNA (10 ng/μL), 0.25 μL of each primer (10 μM), and 4.5 μL of 2× SYBR Green PCR Master Mix (Roche, 4913914001, Mannheim, Germany), using ABI Real-Time PCR systems (Q6 and ViiA7) according to the manufacturer’s instructions. The *OsActin* gene (*LOC_Os03g50885*) was used as the internal control. The relative gene expression levels were calculated by the 2^−∆∆Ct^ method. Each measurement was performed with three biological samples and three replicates for each sample. Relevant primer sequences were listed in Appendix A.

### 4.6. De Novo Assembly of Two Genomes and Sequences Comparison

In order to fine map the candidate genes, the whole genomes of J23B and CR071 were separately sequenced on Illumina and Nanopore (ONT) platforms to capture the target candidate segment sequences. For CR071, 50.3 Gb of ONT data (~135× genome coverage) and 3.8 Gb of Illumina data (~10× genome coverage) were used. For J23B, 10.2 Gb of ONT data (~28× genome coverage) and 5.2 Gb of Illumina data (~14× genome coverage) were used. The Nanopore reads were assembled using Canu [78] for CR071 and using wtdbg2 [79] for J23B. The contigs generated with Canu and wtdbg2 were polished with three rounds of Racon [80] based on Nanopore reads, followed by one round of Pilon [81] based on Illumina short reads. The assembled genomes were used in the subsequent analysis. We then captured the target candidate segment sequences in five genomes using the primer sequences of the two SNP markers and found a 53 kb deletion in J23B compared with CR071 (Appendix A). The sequence comparison of J23B and CR071 was conducted by aligning the ONT reads and Illumina short reads of J23B and CR071 to the three annotated genomes (Zhenshan97, Minghui63 and Nipponbare) using minimap2 [82] and SAMtools [83] to detect the ORF underlying *GLW7.1*.

### 4.7. Haplotype Analysis

The variations in *Ghd7* in 533 accessions were queried from RiceVarMap v2.0 (http://ricevarmap.ncpgr.cn/, accessed on 5 July 2021) with variation IDs (vg0709154754, vg0709154664, vg0709154489, vg0709154469, vg0709154456, vg0709154415, vg0709152671, vg0709152659, vg0709152655, vg0709152479) in the coding region. We then identified nine haplotypes based on the diversity. Relevant data are listed in Appendix A.

### 4.8. Vector Construction and Transformation

For preparing the complementation construct (Com), a 5 kb fragment, which consisted of 2.2 kb promoter and 2.8 kb genomic DNA of *Ghd7*, was amplified from CR071 and then cloned into the plant binary vector pCAMBIA1301. For preparing the CRISPR/Cas9 knockout construct, the sequence (c.512 TGGCCAATGTTGGGGAGAGC) in the second exon was designed as the sgRNA target site. The reverse complement sequence of the target site was inserted into the intermediate vector pER8-Cas9-U6 and then cloned into vector pCXUN-Cas9 [84]. The complementation construct was introduced into NIL-J and the knockout construct was introduced into NIL-C, respectively, by *Agrobacterium tumefaciens* (*EHA105*)-mediated transformation. The transgenic lines were further confirmed by PCR detection and direct sequencing. Relevant primer sequences were listed in Appendix A.

### 4.9. Transcription Activation Assay

The coding sequences of exon1 and exon2 from different allelic *Ghd7* were amplified and combined to generate seven allelic *Ghd7.* These different allelic *Ghd7* were then fused with GAL4 DNA binding domain to generate effectors [53]. The firefly *LUC* gene was used as the reporter to analyze the transcriptional activity. Two allelic *OsMADS1* from Zhenshan97 and Nangyangzhan were fused with GAL4 DNA binding domain to generate the positive controls GALZ and GALN. To explore the downstream target genes, the Hap2 allelic *Ghd7* was fused into the ‘None’ effector vector, while the promoter of candidates (*OsBZR1*, *OsMADS1*, *OsMAPK6*, *OsSPL13* and *OsWRKY53*) was cloned into the ‘190LUC’ reporter vector. The renilla *LUC* gene was used as an internal transformation control. The rice protoplasts prepared from the leaf sheath of Nipponbare and Zhenshan97 seedlings were transfected with different combinations of vectors by PEG-mediated transformation [85]. The firefly luciferase activity was detected after at least 12h using the Dual-Luciferase reporter kit (Promega, E1960, Madison, WI, USA), according to the manufacturer’s protocol. Relevant primer sequences were listed in Appendix A.

### 4.10. Yeast Two-Hybrid Assays

The prey library was derived from young panicles (5–15 cm in length) of Zhenshan97. The coding sequence of the C-terminal of GHD7 (aa. 208–257) was amplified and then cloned into the bait vector pGBKT7 (Clontech, 630443, Mountain View, CA, USA) for yeast two-hybrid screening. Full-length cDNAs of *OsFBK12*, *FZP*, *OsNAC024*, *OsNAC025*, *OsNF-YC12*, *RSR1*, *SNB* and *SLR1* were amplified and then cloned into the prey vector pGADT7 (Clontech, 630442, Mountain View, CA, USA), respectively, for subsequent yeast two-hybrid assays. All procedures were conducted according to the manufacturer’s protocol. Relevant primer sequences are listed in Appendix A.

### 4.11. SFLC Assays

Full-length cDNAs of *Ghd7*, *OsFBK12*, *FZP*, *OsNAC024*, *OsNAC025*, *OsNF-YC12*, *RSR1*, *SNB* and *SLR1* were amplified and then cloned into the nLUC vector (pCAMBIA1300-35S-HA-Nluc-RBS) or cLUC vector (pCAMBIA1300-35S-Cluc-RBS) [9], respectively, for subsequent split firefly luciferase complementation (SFLC) assays. Vectors for testing the protein–protein interactions (such as GHD7-nLUC and FBK12-cLUC), together with the p19 silencing vector, were co-transfected into tobacco (*N. benthamiana*) leaves via *Agrobacterium tumefaciens* (*EHA105*) infiltration. After at least 48 h, injected leaves were sprayed with 5 mM luciferin (Promega, E1605, Madison, WI, USA). The LUC signal was captured using a cooling CCD imaging apparatus (Tanon, Tanon-5200, Shanghai, China). Each assay was repeated at least three times. Relevant primer sequences were listed in Appendix A.

### 4.12. Exogenous GA_3_ and PBZ Treatment of Seedlings

The germinated seeds of NIL-J and NIL-C were grown in a nutrient solution that contained various concentrations of GA_3_ (Sangon Biotech, A600738, Wuhan, China) or 10 μM Paclobutrazol (Sangon Biotech, A630332, Wuhan, China) and incubated at 28 °C under 13 h light/11 h dark conditions. After 10 days, the length of the second leaf sheaths was measured.

### 4.13. Measurement of GA_1_

The shoots of 2-week-old seedlings were sampled, frozen in liquid nitrogen, and ground to fine powder. Tissues weighing 0.1 g were extracted with 1 mL 0.01 M PBS solution at 4 °C for 12 h. After centrifugation (12,000 rpm, 4 °C, 15 min), the supernatant was collected for GA_1_ measurement. Endogenous GA_1_ levels were detected by enzyme-linked immunosorbent assay (ELISA) following the manufacturer’s instructions (Jingmei Biotechnology, JM-110038P2, Yancheng, China).

### 4.14. Statistical Analysis

ANOVA analysis or Student’s *t*-test analysis were conducted using SPSS 22 (SPSS Inc., Chicago, IL, USA).

## Figures and Tables

**Figure 1 ijms-23-08715-f001:**
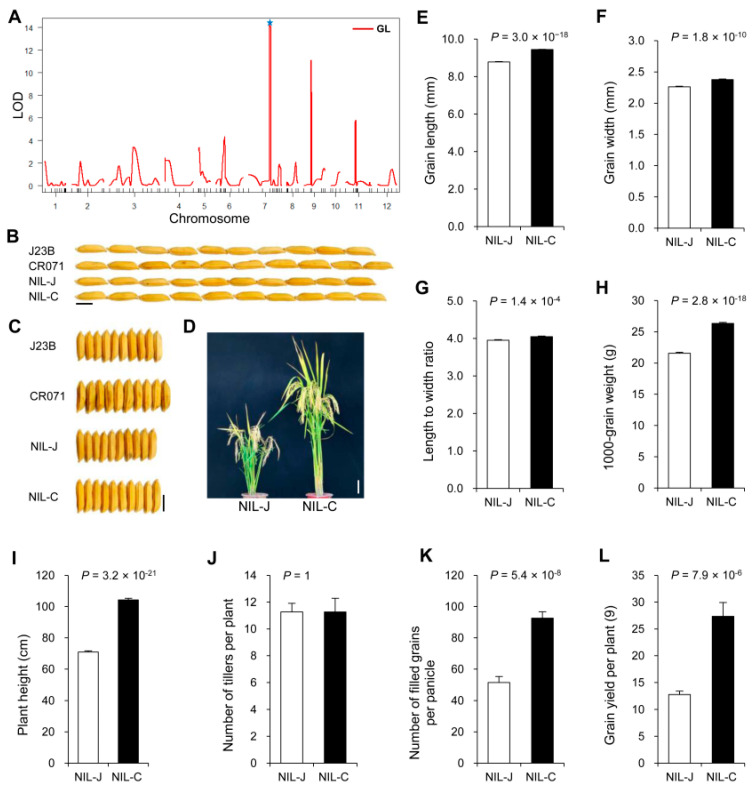
Field trial of *GLW7.1* NIL lines. (**A**) Primary mapping of QTLs for grain length using J23B/CR071 BC_3_F_1_ population (*n* = 238). The star symbol indicates *GLW7.1* locus. (**B**,**C**) Grain morphology. Scale bar: 5 mm. (**D**) The gross morphology of NIL plants. Scale bar: 10 cm. (**E**) Grain length. (**F**) Grain width. (**G**) Length to width ratio. (**H**) 1000-grain weight. (**I**) Plant height. (**J**) Number of tillers per plant. (**K**) Number of filled grains per panicle. (**L**) Grain yield per plant. All phenotypic data in (**E**–**L**) were measured from paddy-grown NIL plants grown under normal cultivation conditions. Data were represented as mean ± s.e.m. (*n* = 15). The Student’s *t*-test was used to produce *p* values.

**Figure 2 ijms-23-08715-f002:**
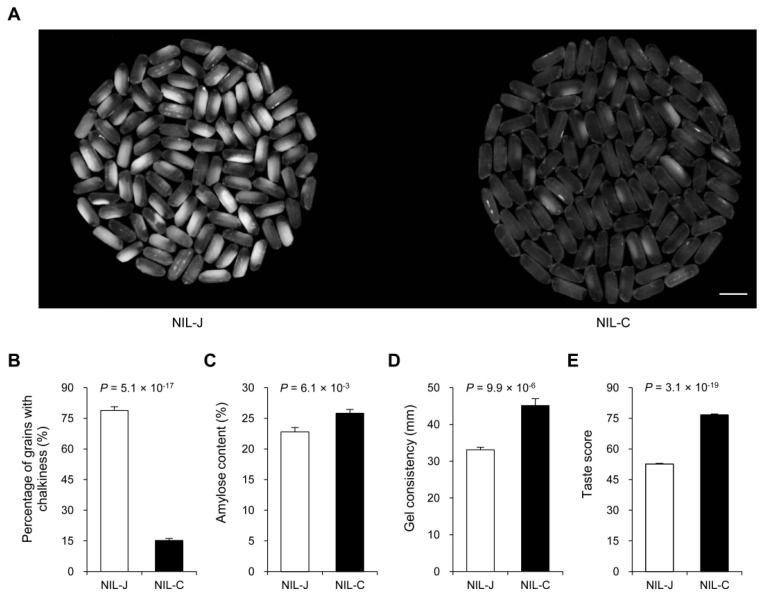
The effects of the *GLW7.1* allele on the physicochemical characteristics of milled rice. (**A**) Comparisons of chalkiness and endosperm transparency of milled rice between the *GLW7.1* NILs (*n* = 100). Scale bar: 1 cm. (**B**) Percentage of grains with chalkiness. (**C**) Amylose content. (**D**) Gel consistency. (**E**) Taste score. All phenotypic data in (**B**–**E**) were measured from paddy-grown NIL plants grown under normal cultivation conditions. Data are represented as mean ± s.e.m. (*n* = 10). The Student’s *t*-test was used to produce *p* values.

**Figure 3 ijms-23-08715-f003:**
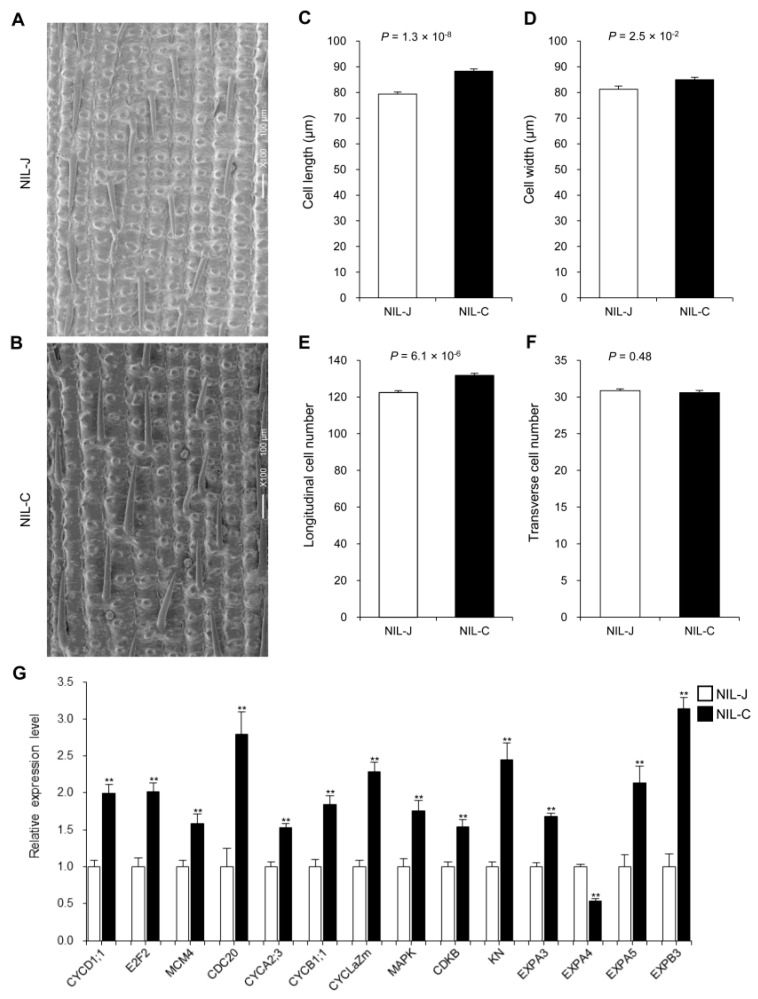
The effect of *GLW7.1* on cell number and cell size. (**A**–**F**) Scanning electron microscope analysis. Scale bar: 100 μm. (**A**) Outer epidermal cells of NIL-J. (**B**) Outer epidermal cells of NIL-C. (**C**) Average cell length. (**D**) Average cell width. (**E**) Total number of longitudinal cells. (**F**) Total number of transverse cells. Data are represented as mean ± s.e.m. (*n* = 15). The Student’s *t*-test was used to produce *p* values. (**G**) Relative expression level of 10 cell cycle-related genes and 4 cell expansion genes in young panicles (8–10 cm in length) of NIL-J and NIL-C. *OsActin* (*LOC_Os03g50885*) was used as the control and the values of expression level in NIL-J were set to 1. Data are represented as mean ± s.e.m. (*n* = 9). The Student’s *t*-test was used to produce *p* values (** indicates *p* < 0.01).

**Figure 4 ijms-23-08715-f004:**
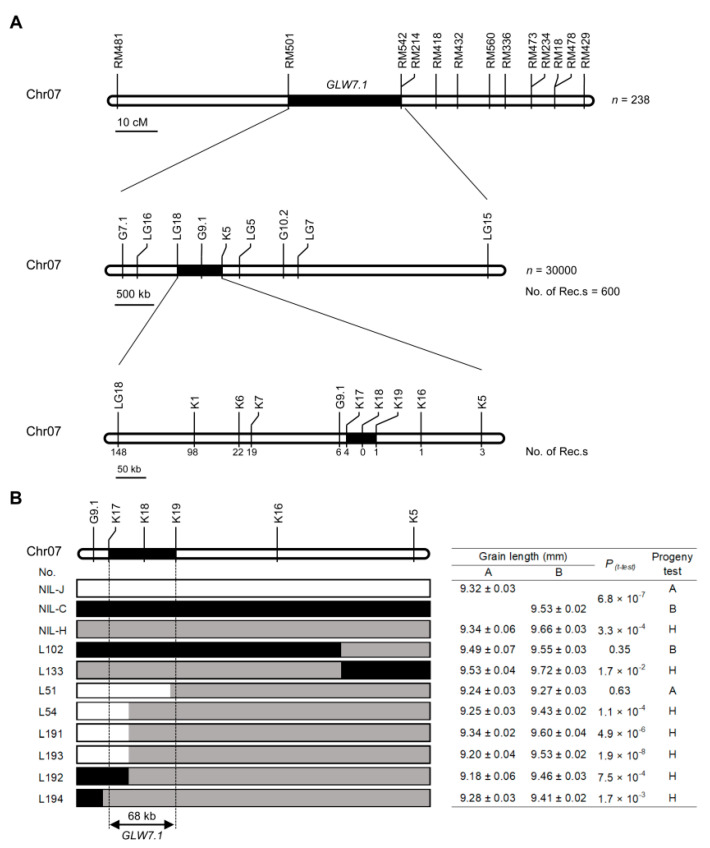
Map-based cloning of *GLW7.1.* (**A**) Fine mapping of the *GLW7.1* using 30,000 BC_5_F_2_ segregants. Numbers below the line indicate the number of recombinants between *GLW7.1* and the marker shown. (**B**) Genotypes and phenotypes of the recombinants. Grain length (mean ± s.e.m.) of three near-isogenic lines (NIL), and recombinant BC_5_F_3_ lines (L102, L133, L51, L54, L191, L193, L192, L194). White bars represent chromosomal segments for J23B homozygote (progeny test named as A), black for CR071 homozygote (progeny test named as B), and grey for heterozygotes (progeny test named as H). Homozygous progenies from each line were harvested to compare phenotypic differences. The Student’s *t*-test was used to produce *p* values.

**Figure 5 ijms-23-08715-f005:**
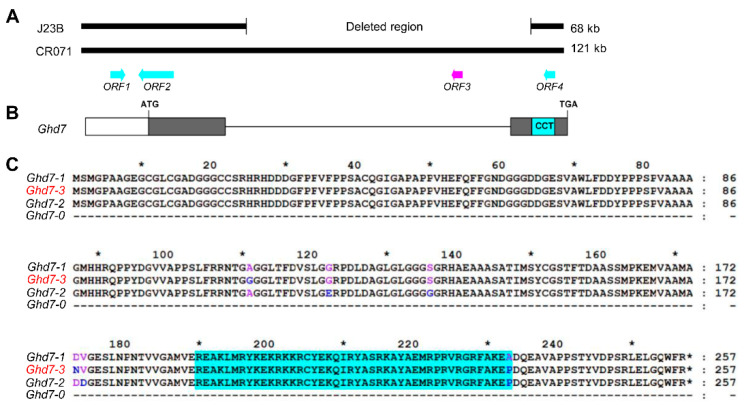
Candidate gene of *GLW7.1*. (**A**) A deletion of 53 kb was detected in J23B compared with CR071, which included the candidate gene of *GLW7.1* (the arrow symbol in magenta), while no variation was detected in the protein-coding regions of the other three ORFs (arrow symbols in cyan) and their corresponding promoter regions. (**B**) *ORF3* encodes the CCT motif family protein, GHD7. CCT domain is indicated in the cyan box. (**C**) The protein sequences of GHD7 for four *Ghd7* alleles. CR071 carried *Ghd7-3* allele of *Ghd7* compared with Minghui 63 (*Ghd7-1*), Nipponbare (*Ghd7-2*) and J23B (*Ghd7-0*). CCT domain (aa. 190–233) is indicated with cyan background. Polymorphic amino acids are indicated by different colors using the amino acids sequence of *Ghd7-1* allele as reference. Asterisks above amino acid sequences indicate positions of amino acids (10, 30, 50, 70, 90, 110, 130, 150, 170, 190, 210, 230 and 250) and asterisks within amino acid sequences indicate stop codons.

**Figure 6 ijms-23-08715-f006:**
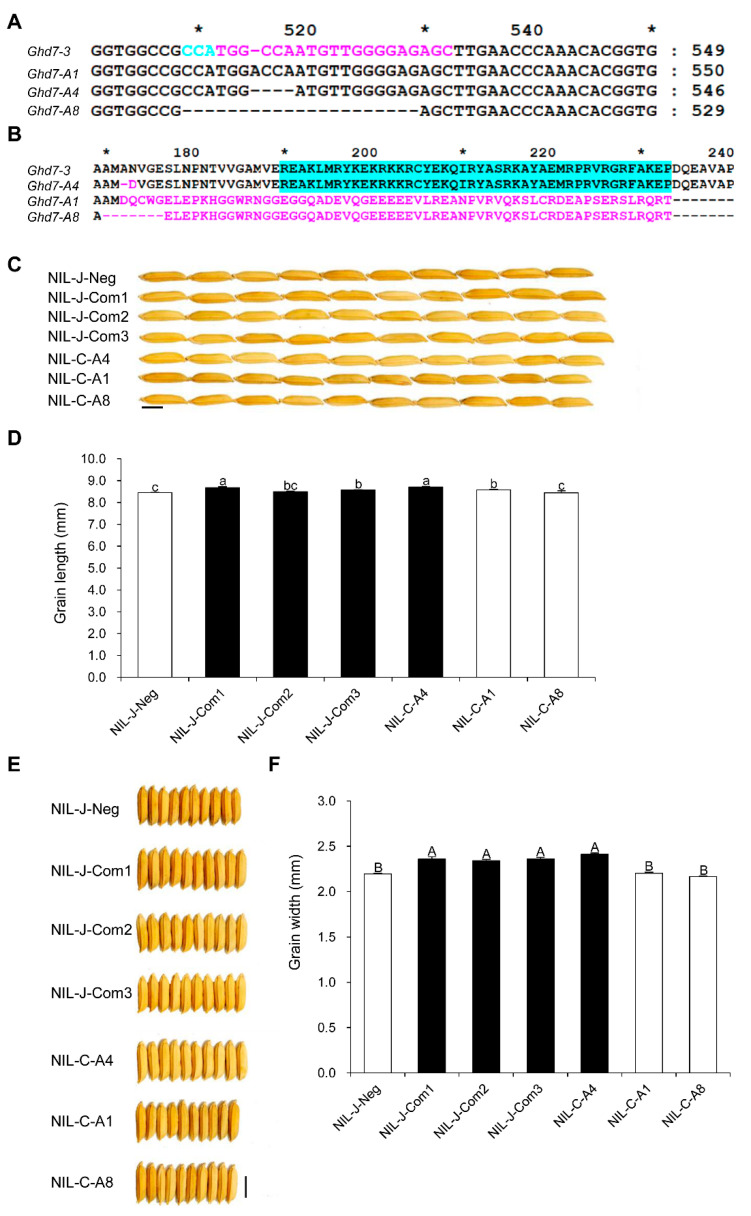
Transgenic validation of *GLW7.1*. (**A**,**B**) Gene editing targeting the upstream of CCT domain was conducted using CRISPR-Cas9 system. Compared with the amino acid change mutation (AN → D) caused by mutated allele A4, the mutated allele A1 with 1 bp insertion and A8 with 20 bp deletion resulted in frameshift mutations, which caused the loss of the CCT domain in the GHD7 protein. (**A**) The sgRNA target site is shown in magenta, the PAM sequence is shown in cyan, and asterisks above nucleotide sequences indicate positions of nucleotides (510, 530 and 550). (**B**) CCT domain is indicated in cyan background, polymorphic amino acids are indicated in magenta, and asterisks above amino acid sequences indicate positions of amino acids (170, 190, 210 and 230). (**C**,**E**) Grains morphology of transgenic lines. Scale bar: 0.5 cm. (**D**) Grain length of transgenic lines. (**F**) Grain width of transgenic lines. All phenotypic data were measured from paddy-grown transgenic lines grown under normal cultivation conditions. Data are represented as mean ± s.e.m. (*n* = 27 for NIL-J-Neg, 15 for NIL-J-Com1, 22 for NIL-J-Com2, 20 for NIL-J-Com3, 10 for NIL-C-A4, 14 for NIL-C-A1 and 4 for NIL-C-A8) and Duncan’s multiple range tests were used to conduct statistical analysis (a, b and c indicate *p* < 0.05; A and B indicate *p* < 0.01).

**Figure 7 ijms-23-08715-f007:**
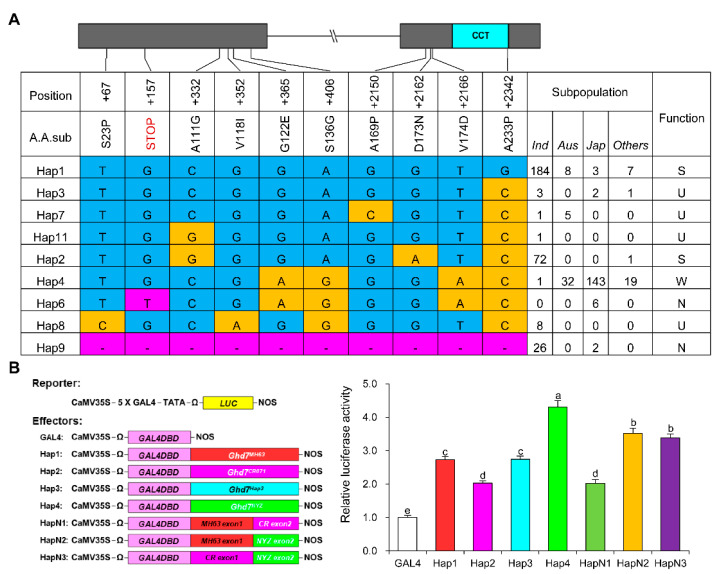
Haplotype analysis of *Ghd7*. (**A**) The nonsynonymous polymorphisms in the *Ghd7* CDS region that cause changes in the amino acid sequence of 525 cultivars. CCT domain is indicated in the cyan box. Polymorphic nucleotides that cause amino acid substitutions are indicated in yellow using the amino acids sequence of Hap1 as reference, and the nucleotides that cause frame-shift mutation or absence of the gene region are indicated in magenta. S, U, W and N represent strong functional, unknown functional, weak functional and nonfunctional alleles, respectively. (**B**) The transactivation activity of different allelic GHD7 proteins. Six allelic GHD7 were fused to the GAL4 DNA-binding domain (GAL4DBD). The relative activity of firefly luciferase (LUC) under control of the 5×GAL4-binding element was measured. Renilla luciferase (REN) activity was used as internal control. Data are represented as mean ± s.e.m. (*n* = 10) and Duncan’s multiple range tests were used to conduct statistical analysis (a, b, c, d and e indicate *p* < 0.05).

**Figure 8 ijms-23-08715-f008:**
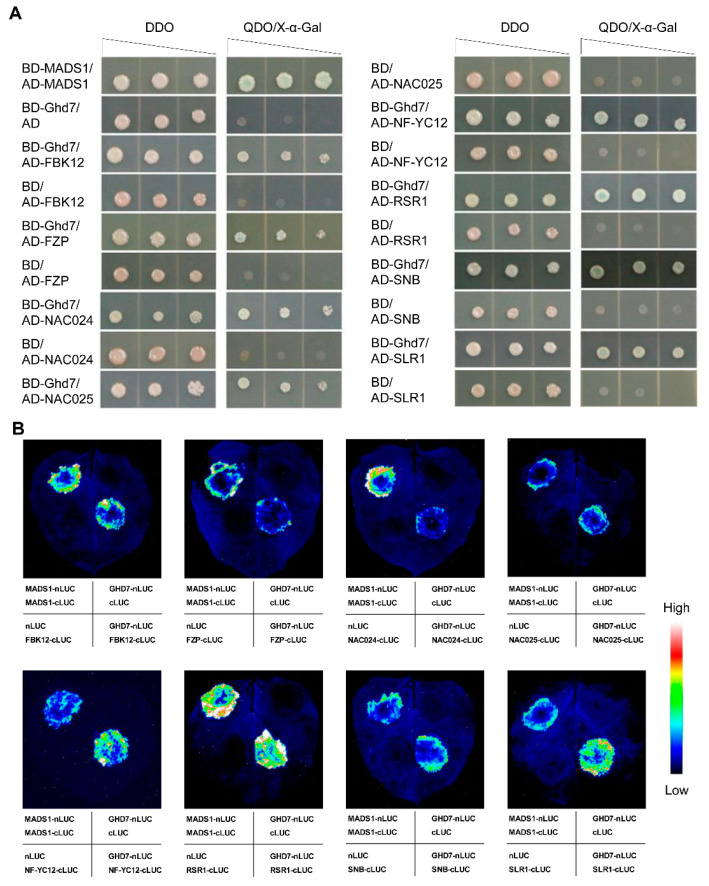
Eight candidate proteins that interact with GHD7. (**A**) Yeast two-hybrid assays. Serial dilutions of 10^3^–10 transformed yeast cells were spotted on the control medium DDO (SD/-Trp/-Leu) and selective medium QDO (SD/-Trp/-Leu/-His/-Ade). The protein self-dimerization of OsMADS1 was used as positive controls. Co-transformed empty vectors pGADT7 (AD) and pGBKT7 (BD) were used as negative controls. (**B**) Split firefly luciferase complementation (SFLC) assays. nLUC-tagged GHD7 was co-transformed into tobacco leaves along with the cLUC-targeted candidate proteins. The protein self-dimerization of OsMADS1 was used as positive controls. Co-transformed empty vectors nLUC and cLUC were used as negative controls.

**Figure 9 ijms-23-08715-f009:**
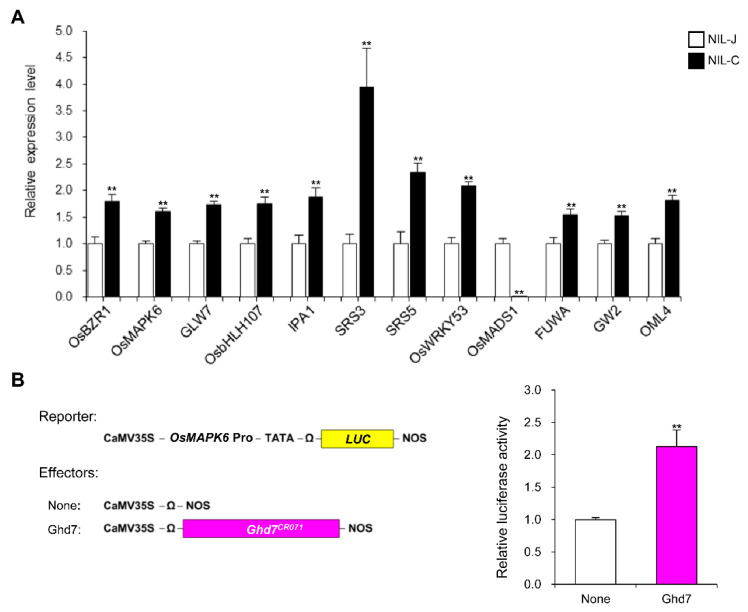
*GLW7.1* regulates the expression of grain-size genes. (**A**) Relative expression level of 12 grain-size-related genes in young panicles (8–10 cm) of NIL-J and NIL-C. *OsActin* (*LOC_Os03g50885*) was used as the control and the values of expression level in NIL-J were set to 1. Data were represented as mean ± s.e.m. (*n* = 9). The Student’s *t*-test was used to produce *p* values (** indicates *p* < 0.01). (**B**) GHD7 induces the LUC activity driven by *OsMAPK6* promoter. Hap2 allelic GHD7 were overexpressed as effectors. The relative activity of firefly luciferase (LUC) under control of the promoter region of *OsMAPK6* was measured. Renilla luciferase (rLUC) activity was used as internal control. Data are represented as mean ± s.e.m. (*n* ≥ 5). The Student’s *t*-test was used to produce *p* values (** indicates *p* < 0.01).

**Figure 10 ijms-23-08715-f010:**
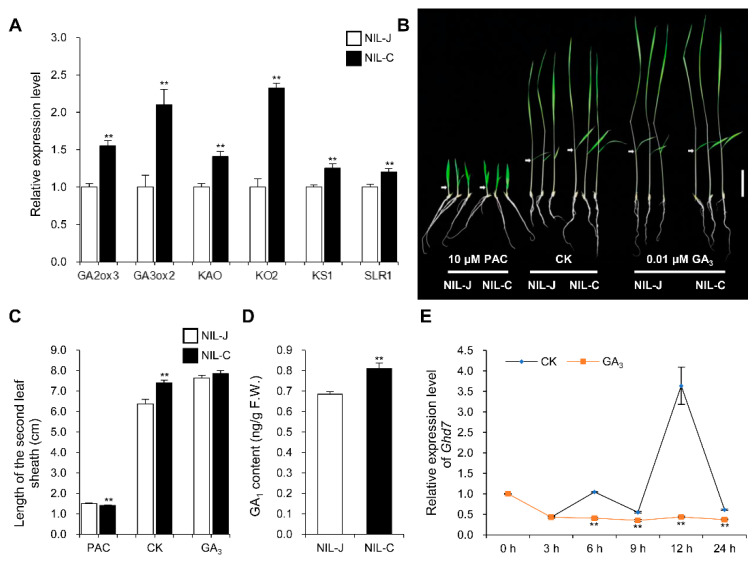
GLW7.1 participates in the biosynthesis of GA. (**A**) Relative expression level of 6 GA related genes in young panicles (8–10 cm) of NIL-J and NIL-C. *OsActin* (*LOC_Os03g50885*) was used as the control and the values of expression level in NIL-J were set to 1. Data were represented as mean ± s.e.m. (*n* = 9). Student’s *t*-test was used to produce *p* values (** indicates *p* < 0.01). (**B**) Seedling phenotype of NIL-J was rescued by 0.01 μM GA_3_. The germinated seeds were grown in the nutrient solution that contained 0.01 μM GA_3_ or 10 μM paclobutrazol (PAC) and incubated at 28 °C under 13 h light/11 h dark conditions. CK, nutrient solution without any exogenous hormones. After 10 days, the seedlings were photographed. Scale bar: 5 cm. Arrow symbols indicate the second leaf sheaths. (**C**) The length of the second leaf sheaths were measured after treatment. Data are represented as mean ± s.e.m. (*n* ≥ 15). Student’s *t*-test was used to produce *p* values (** indicates *p* < 0.01). (**D**) Levels of endogenous GA1 in NIL-J and NIL-C 2-week-old seedlings grown in nutrient solution without treatment. Data are represented as mean ± s.e.m. (*n* = 4). Student’s *t*-test was used to produce *p* values (** indicates *p* < 0.01). (**E**) Expression pattern of *Ghd7* under GA_3_ treatment. The Zhonghua11 germinated seeds were grown in the nutrient solution. Two weeks later, half of the seedlings were moved to the nutrient solution that contained 50 μM GA_3_, while the other half were moved to the nutrient solution without treatment as control. Relative expression level of *Ghd7* were detected. *OsActin* was used as the control and the values of expression level at 0 h were set to 1. Data were represented as mean ± s.e.m. (*n* ≥ 3). Student’s *t*-test was used to produce *p* values (** indicates *p* < 0.01).

## Data Availability

The whole-genome sequencing data in this paper can be found in the NCBI database under the following accession numbers: The whole-genome resequencing of J23B and CR071 (PRJNA791417), Illumina reads of J23B (SRR17299467), Nanopore reads of J23B (SRR17299468), Illumina reads of CR071 (SRR17299469), Nanopore reads of J23B (SRR17299470). Gene sequence in this paper can be found in the Rice Genome Annotation Project Database (http://rice.uga.edu/): *Ghd7* (*LOC_Os07g15770*).

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
