# Peer review of "GLW7.1, a Strong Functional Allele of Ghd7, Enhances Grain Size in Rice"

_ijms, 2022, doi:10.3390/ijms23158715_

Round 1

Reviewer 1 Report

The authors present different sets of data that grain size in rice as as other traits are strongly coupled to the transcription factor GHD7.

It´s remarkable that the authors utilized a number of genetical and molecular biological methods to come to this conclusion and and support their arguments. The studies are properly done and well documented. Therefore, I would just have some minor comments/suggestions:

- The sentence in the abstract going from line 12-15 is a bit unclear to me. Please reformulate.

- The abbreviation SFLC (line 17) first of all confused me. Please write out at that position once.

- In the introduction a number of transcription factors and other genes in relation to rice growth traits are mentioned which makes the impression of sheer listing. I would suggest to condense that part and integrate a paragraph about the studies on GHD7 done before.

Author Response

Dear Reviewer:

Thank you for your comments concerning our manuscript entitled “GLW7.1, a strong functional allele of Ghd7, enhances grain size in rice(ID: ijms-1832922). Those comments are all valuable and very helpful for revising and improving our manuscript, as well as the important guiding significance to our researches. We have studied comments carefully and have made correction which we hope meet with approval. Revised portion are marked in red in the revised manuscript. The replies to the reviewer’s comments are listed below.

  1. The sentence in the abstract going from line 12-15 is a bit unclear to me. Please reformulate.

[Response 1]

Thanks for your comment. We have made changes as suggested in the revised manuscript (line 12-15: “Scanning electron microscopy analysis and expression analysis revealed that GLW7.1 promotes the transcription of several cell division and expansion genes, further resulting in larger cell size and increased cell number, and finally enhancing the grain size as well as grain weight.”).

  1. The abbreviation SFLC (line 17) first of all confused me. Please write out at that position once.

[Response 2]

Thanks for your comment. SFLC is the abbreviation of split firefly luciferase complementation. We have made changes as suggested in the revised manuscript (line 17: “split firefly luciferase complementation”).

  1. In the introduction a number of transcription factors and other genes in relation to rice growth traits are mentioned which makes the impression of sheer listing. I would suggest to condense that part and integrate a paragraph about the studies on GHD7 done before.

[Response 3]

Thanks for your suggestion. In this section, we summarize the recent advances in the study of grain size genes to describe the major regulatory networks of rice grain size, including G protein signaling pathway, the ubiquitin-proteasome pathway, mitogen-activated protein kinase (MAPK) signaling pathway, phytohormone signaling and homeostasis, and transcriptional regulators, some of which have subsequently been shown to be involved in GLW7.1-mediated regulation of grain size. Therefore, we did not condense that part in the revised manuscript.

Although Ghd7 is a cloned gene, previous studies have not involved the relationship between Ghd7 and grain development, and its homologous proteins have not been reported to be involved in the regulation of grain size too. We do not think it is appropriate to describe some of the previous studies that are not closely related to our work in a single paragraph. Moreover, the previous research articles [1,2] on Ghd7 also did not introduce in detail the studies on GHD7 done before which were not closely related to their works. However, we have added a paragraph about the studies on GHD7 done before as suggested in the revised manuscript (line 84-88: “Grain number, plant height, and heading date 7 (Ghd7) was first reported as a major regulator of heading date, and improved yield by increasing grain number [3]. Subsequent studies revealed that it participated in a variety of other developmental processes, such as stress responses, seed germination and nitrogen utilization [1,2,4].”).

References

  1. Wang, Q.; Su, Q.; Nian, J.; Zhang, J.; Guo, M.; Dong, G.; Hu, J.; Wang, R.; Wei, C.; Li, G.; et al. The Ghd7 transcription factor represses ARE1 expression to enhance nitrogen utilization and grain yield in rice. Mol. Plant 2021, 14, 1012-1023, doi:10.1016/j.molp.2021.04.012.
  2. Hu, Y.; Song, S.; Weng, X.; You, A.; Xing, Y. The heading-date gene Ghd7 inhibits seed germination by modulating the balance between abscisic acid and gibberellins. Crop J. 2021, 9, 297-304, doi:10.1016/j.cj.2020.09.004.
  3. Xue, W.; Xing, Y.; Weng, X.; Zhao, Y.; Tang, W.; Wang, L.; Zhou, H.; Yu, S.; Xu, C.; Li, X.; et al. Natural variation in Ghd7 is an important regulator of heading date and yield potential in rice. Nat. Genet. 2008, 40, 761-767, doi:10.1038/ng.143.
  4. Weng, X.; Wang, L.; Wang, J.; Hu, Y.; Du, H.; Xu, C.; Xing, Y.; Li, X.; Xiao, J.; Zhang, Q. Grain number, plant height, and heading date7 is a central regulator of growth, development, and stress response. Plant Physiol. 2014, 164, 735-747, doi:10.1104/pp.113.231308.

Reviewer 2 Report

In this article Liu et al. described the identification of a new gene involved in the determinism of grain size in rice.

Using an impressive combination of quantitative genetics, molecular biology and imaging techniques, they identified QTLs for grain size and underlying candidate genes. The most promising one was found to exist in a panel of rice accessions with sequences variations that correlated with grain size. Mutants and complementation assays confirmed that the Ghd7 gene was responsible for the observed grain size variation. The mechanisms by which the identified gene regulates grain size was investigated. The nature of the gene suggested a transcription factor activity. Therefore, the authors performed assays for transcription activation, protein-protein interactions and targeted gene expression analysis, they identified partners and downstream genes. NILs were identified for Ghd7 and were analysed including in terms of food quality.

To my opinion the article is clear and well written.

The number of repetitions and the controls are globally correct.

Several points require clarifications:

Quality traits. Line 118-120 and + discussion 396-400: the authors should better explain why they studied chalkiness and other quality traits and Taste Score in rice and add references e.g. Cheng et al. 2019  doi.org/10.1016/j.jcs.2019.02.009; Champagne et al. 1996 Quality Evaluation of U.S. Medium-Grain Rice Cereal Chemistry. Discussion. Any hypothesis concerning why this gene enhances the studied quality traits?

 Cytology: cells on the surface of the lemma. Number counted manually and cell length and width by image J. On the micrographs in figure 3 the cell longitudinal borders cannot be seen. Can the authors better explain how they use Image J to monitor cell length. The authors should also better justify why they looked at cell number and size at the lemma surface (why not the palea for instance), at least give a reference to show the link with grain size (e.g; Shomura 2008; Li and Li 2015).

“the number of longitudinal cells were statistically different” Figure 3 L 132-134 and 566-569: I don’t understand what the authors call longitudinal cells and transverse cells. Do they mean number of cells along the grain longitudinal axis and number of cells along the grain transverse axis (the second would be the equivalent of number of cell rows?).

 Gene expression studies. L 137-142 and 320-325: It is surprising that considering all the experiments carried out for this article the authors chose to study the expression of a few selected genes by qRT-PCR rather than RNA seq. RNAseq would have been more robust to identify altered pathway. Moreover, the MIQE guidelines for gene expression studies by qRT-PCR (Bustin et al. 2009) recommend the use of several reference genes, here the authors used only one actin gene. It might be sufficient as the reference gene is indicated in the figure legend but the authors should add the gene id in the legend since there are several actin genes in the rice genome. “The relative gene expression levels were calculated by the 2-ΔΔCt method.” Did the authors check that the amplification efficiency of all the primer pairs indicated in Table S2 were comparable.

 Minor points

Many abbreviations require to be defined e.g. Line 38 Define GS, L 46 define GW, L 48 define CLG, L 50 define WTG, L 52 Define LG1/OsUBP15, L 56 define GSN1, L 64 define BR, L 76 Define CCT motif and more…

L 139 “3 cell expansion genes” / L 148 “4 cell expansion genes in young panicles (8–10 cm in length)”. Which one is correct?

L 606-607 give a reference for CRISPR-Cas9

L 220, 531, 604, 610, table S5: complementation construct/test/experiment rather than complementary or commentary construct/experiment

L 222 complemented lines rather than complemental lines

Fig S6: the figure legend is written in red

Author Response

Dear Reviewer:

Thank you for your comments concerning our manuscript entitled “GLW7.1, a strong functional allele of Ghd7, enhances grain size in rice(ID: ijms-1832922). Those comments are all valuable and very helpful for revising and improving our manuscript, as well as the important guiding significance to our researches. We have studied comments carefully and have made correction which we hope meet with approval. Revised portion are marked in red in the revised manuscript. The replies to the reviewer’s comments are listed below.  

  1. Quality traits. Line 118-120 and + discussion 396-400: the authors should better explain why they studied chalkiness and other quality traits and Taste Score in rice and add references e.g. Cheng et al. 2019  doi.org/10.1016/j.jcs.2019.02.009; Champagne et al. 1996 Quality Evaluation of U.S. Medium-Grain Rice Cereal Chemistry. Discussion. Any hypothesis concerning why this gene enhances the studied quality traits?

[Response 1]

Thanks for your suggestion. We have made changes as suggested in the revised manuscript (Line 131-136: “The dominant GLW7.1 locus with yield-increasing effect has a good advantage in hybrid rice breeding. Considering that the simultaneous increase in rice grain length and width is usually accompanied by a decrease in rice quality, to further evaluate the prospects of GLW7.1 in rice breeding, we then examined rice quality traits among NILs, including percentage of grains with chalkiness, amylose content, gel consistency, and taste value.”) to explain why we studied quality traits. We have also added the website of the NY/T 593-2013 standard (Line 565) and the reference “Champagne et al. 1996 Quality Evaluation of U.S. Medium-Grain Rice Cereal Chemistry.” (Line 566).

NIL-C shows increased grain size (Figure 1B, C, E, F) and better quality (Figure 2), which looks contradictory. However, we noticed the protein-protein interaction between GHD7 and NF-YC12 (Figure 8). The loss-of-function mutants of OsNF-YC12 showed a decrease in grain length and width, an increase in chalkiness and a decrease in amylose starch content [1,2], which were similar to NIL-J. Further investigation revealed that NF-YC12 could directly activate the expression of several genes related to accumulation of storage-substance, such as SUT1, GS1;3 and FlO6, to regulate rice quality [2]. We hypothesized that GHD7 may interact with NF-YC12 to further promote the expression of these downstream genes, thereby increasing grain size and simultaneously improving quality.

  1. Cytology: cells on the surface of the lemma. Number counted manually and cell length and width by image J. On the micrographs in figure 3 the cell longitudinal borders cannot be seen. Can the authors better explain how they use Image J to monitor cell length. The authors should also better justify why they looked at cell number and size at the lemma surface (why not the palea for instance), at least give a reference to show the link with grain size (e.g; Shomura 2008; Li and Li 2015). “the number of longitudinal cells were statistically different” Figure 3 L 132-134 and 566-569: I don’t understand what the authors call longitudinal cells and transverse cells. Do they mean number of cells along the grain longitudinal axis and number of cells along the grain transverse axis (the second would be the equivalent of number of cell rows?).

[Response 2]

Thanks for your suggestion. To monitor cell length, the bipeaked tubercles on the surface of the outer epidermal cells of the lemma were taken as the measuring points, and the average cell length was obtained by measuring the distance between two points in a column (Figure R1).

We have added the reference to show the link of glume with grain size in the revised manuscript (Line 148-149: “The glume, including lemma and palea, determines the upper limit of grain size [3-6], and its size is determined by cell number and cell size.”). Previous studies have shown that the number and size of cells in palea and lemma have almost the same trend when the grain size altered [4-6]. Since the lemma was larger and easier to observe and count, we did not investigate the palea further due to the consideration of time and physical costs.

As you understand, longitudinal cells and transverse cells mean cells along the grain longitudinal axis (the grain-length direction) and cells along the grain transverse axis (the grain-width direction).

Figure R1. Outer epidermal cells of NIL-J. The bipeaked tubercles were indicated by arrow symbols. The distance (D) between two tubercles were obtained by image J. Average cell length = D / (bipeaked tubercles number – 1). At least three sets of data were measured for each image, and at least five photos were taken for each line.

  1. Gene expression studies. L 137-142 and 320-325: It is surprising that considering all the experiments carried out for this article the authors chose to study the expression of a few selected genes by qRT-PCR rather than RNA seq. RNAseq would have been more robust to identify altered pathway. Moreover, the MIQE guidelines for gene expression studies by qRT-PCR (Bustin et al. 2009) recommend the use of several reference genes, here the authors used only one actin gene. It might be sufficient as the reference gene is indicated in the figure legend but the authors should add the gene id in the legend since there are several actin genes in the rice genome. “The relative gene expression levels were calculated by the 2-ΔΔCt method.” Did the authors check that the amplification efficiency of all the primer pairs indicated in Table S2 were comparable?

[Response 3]

Thanks for your suggestion. Compared with the qRT-PCR, RNA-seq could identify the altered pathway more efficiently. In our study, we only wanted to investigate the effect of GLW7.1 on the expression of cell cycle and expansion genes, and whether GLW7.1 could control grain size by regulating cloned genes related to grain size. Our laboratory has previously collected 43 genes related to cell cycle and expansion and 63 genes related to grain size, so it is more convenient and direct for us to detect the expression differences of these genes by qRT-PCR.

As suggested, we used another reference gene, Ubiquitin (LOC_Os03g13170), which is often used in studies on rice grain size [7-9]. After replacing the reference gene, the results of qRT-PCR (Figure R2) were basically consistent with those in the manuscript.

We provided the gene id in the “Materials and Methods” section of the original manuscript (Line 575-576: “The OsActin gene (LOC_Os03g50885) was used as the internal control.”). According to the suggestion, we have added the gene id “(LOC_Os03g50885)” in the legend in the revised manuscript (Line 169, Line 357, Line 392).

Because these primer pairs were collected directly from published articles, we did not previously check the amplification efficiency of these primer pairs. In order to ensure the accuracy of our qRT-PCR results, we used LinRegPCR software to analyze the amplification data of qRT-PCR to calculate the amplification efficiency of all the primer pairs [10] (Table R1). Among these primer pairs, we found that the low amplification efficiency of the primer pair for detecting SMG11 gene expression was not suitable for subsequent analysis, so we removed SMG11 from the revised manuscript (Line 354-355: Figure 9).

 Figure R2. Expression analysis in young panicles (8–10 cm in length) of NIL-J and NIL-C. (A) Relative expression level of 10 cell cycle-related genes and 4 cell expansion genes (B) Relative expression level of 13 grain size related genes. (C) Relative expression level of 6 GA related genes. Ubiquitin (LOC_Os03g13170) was used as the control and the values of expression level in NIL-J were set to 1. Data were represented as mean ± s.e.m. (n = 9). Student’s t-test was used to produce P values (** indicates P < 0.01).

Table R1. Amplification efficiency of all the qRT-PCR primer pairs.

Figure 9. GLW7.1 regulates the expression of grain-size genes.

Minor points

  1. Many abbreviations require to be defined e.g. Line 38 Define GS, L 46 define GW, L 48 define CLG, L 50 define WTG, L 52 Define LG1/OsUBP15, L 56 define GSN1, L 64 define BR, L 76 Define CCT motif and more…

[Response 1]

Thanks for your suggestion. We have defined these abbreviations the revised manuscript, and some abbreviations that were not easy to define were replaced by other common names for the gene. Changes are as following.

Line 38 GS3 > GRAIN SIZE 3 (GS3),

L 39 DEP1 > L 40 DENSE AND ERECT PANICLE 1 (DEP1),

L 46 GW2 > L47 GRAIN WIDTH 2 (GW2),

L 48 CLG1 > L49 CHANG LI GENG 1 (CLG1),

L 50 WTG1 > L 51-52 WIDE AND THICK GRAIN 1 (WTG1),

L 52 LG1/OsUBP15 > L 54 LARGE GRAIN 1 (LG1),

L 55 GSN1 > L 58-59 GRAIN SIZE AND NUMBER 1 (GSN1),

L 63 BR > L66 brassinosteroid (BR),

L 63 GW5, GL2 and qGL3/OsPPKL1 > L 66-67 GRAIN WIDTH 5 (GW5), GRAIN LENGTH 2 (GL2) and GRAIN LENGTH 3.1 (GL3.1),

L 65 TGW6, BG1 and qTGW3 > L 68-70 THOUSAND GRAIN WEIGHT 6 (TGW6), BIG GRAIN 1 (BG1) and THOUSAND GRAIN WEIGHT 3 (TGW3),

L 65 GA > L 70 gibberellic acid (GA),

L 66 GDD1, SRS3/SGL and SGD2 > L 71-72 GIBBERELLIN-DEFICIENT DWARF 1 (GDD1), SMALL AND ROUND SEED 3 (SRS3) and SMALL GRAIN AND DWARF 2 (SGD2),

L 68 GLW7 > L 74-75 GRAIN LENGTH AND WIEIGHT 7 (GLW7),

L 68 GW8 > L 75 GRAIN WIDTH 8 (GW8),

L 69 An-1 > L 76 Awn-1 (An-1),

L 70 SMOS1, OsSNB/SSH1, FZP > L 77-78 SMALL ORGAN SIZE1 (SMOS1), SUPERNUMERARY BRACT (SNB), FRIZZY PANICLE (FZP),

L 71 GS9, SG6, GL4 > L 79 GRAIN SHAPE 9 (GS9), SHORT GRAIN6 (SG6), GRAIN LENGTH 4 (GL4),

L 75 CCT > L 83-84 CCT (CONSTANS, CONSTANS-LIKE, and TIMING OF CHLOROPHYLL A/B BINDING1),

L 157 SSR and KASP > L 178-179 simple sequence repeats (SSR) and kompetitive allele-specific PCR (KASP),

L 302 RSR1 > L 323 RICE STARCH REGULATOR 1 (RSR1),

L 303 SLR1 > L324 SLENDER RICE 1 (SLR1),

L 324 IPA1 > L345 IDEAL PLANT ARCHITECTURE 1 (IPA1),

L 325 SRS5 > L 346 SMALL AND ROUND SEED 5 (SRS5),

L 464 D11 > L 486 DWARF 11 (D11).

  1. L 139 “3 cell expansion genes” / L 148 “4 cell expansion genes in young panicles (8–10 cm in length)”. Which one is correct?

[Response 2]

Thanks for your suggestion. They are both correct. Among the differentially expressed cell expansion genes, three cell expansion genes (EXPA3, EXPA5 and EXPB3) were significantly up-regulated and one gene (EXPA4) was down-regulated. We hypothesized that the down-regulation of the EXPA4 might be due to an unclear feedback regulatory mechanism. For the sake of data authenticity, we retain the expression of EXPA4 in Figure 3G, which resulted in one more gene in the Figure 3G than described in the corresponding result section.

  1. L 606-607 give a reference for CRISPR-Cas9

[Response 3]

Thanks for your suggestion. We have added the reference (He et al. 2017) in the revised manuscript (Line 629).

  1. L 220, 531, 604, 610, table S5: complementation construct/test/experiment rather than complementary or commentary construct/experiment

[Response 4]

Thanks for your suggestion. We apologize for the careless spelling mistakes in the preparation of the manuscript, and we have made changes as suggested in the revised manuscript. Changes are as following.

L 219 complementary experiment > L 240 complementation experiment,

L 530 commentary test > L 550 complementation test,

L 603 and L 609 complementary construct > L 623 and L 629 complementation construct,

Table S5 complemental lines > complemented lines.

  1. L 222 complemented lines rather than complemental lines

[Response 5]

Thanks for your suggestion. We have made changes as suggested in the revised manuscript (L 242-243: “The grains produced from the complemented lines Com1, Com2 and Com3 were larger than those from negative transgenic plants NIL-J-Neg (Figure 6C-F).”).

  1. Fig S6: the figure legend is written in red

[Response 6]

Thanks for your suggestion. Once again, we apologize for our carelessness, and we have changed the color in the revised manuscript.

References

  1. Bello, B.K.; Hou, Y.; Zhao, J.; Jiao, G.; Wu, Y.; Li, Z.; Wang, Y.; Tong, X.; Wang, W.; Yuan, W.; et al. NF-YB1-YC12-bHLH144 complex directly activates Wx to regulate grain quality in rice (Oryza sativa L.). Plant Biotechnol. J. 2019, 17, 1222-1235, doi:10.1111/pbi.13048.
  2. Xiong, Y.; Ren, Y.; Li, W.; Wu, F.; Yang, W.; Huang, X.; Yao, J. NF-YC12 is a key multi-functional regulator of accumulation of seed storage substances in rice. J. Exp. Bot. 2019, 70, 3765-3780, doi:10.1093/jxb/erz168.
  3. Li, N.; Li, Y. Maternal control of seed size in plants. J. Exp. Bot. 2015, 66, 1087-1097, doi:10.1093/jxb/eru549.
  4. Hu, J.; Wang, Y.; Fang, Y.; Zeng, L.; Xu, J.; Yu, H.; Shi, Z.; Pan, J.; Zhang, D.; Kang, S.; et al. A rare allele of GS2 enhances grain size and grain yield in rice. Mol. Plant 2015, 8, 1455-1465, doi:10.1016/j.molp.2015.07.002.
  5. Shomura, A.; Izawa, T.; Ebana, K.; Ebitani, T.; Kanegae, H.; Konishi, S.; Yano, M. Deletion in a gene associated with grain size increased yields during rice domestication. Nat. Genet. 2008, 40, 1023-1028, doi:10.1038/ng.169.
  6. Zhao, D.S.; Li, Q.F.; Zhang, C.Q.; Zhang, C.; Yang, Q.Q.; Pan, L.X.; Ren, X.Y.; Lu, J.; Gu, M.H.; Liu, Q.Q. GS9 acts as a transcriptional activator to regulate rice grain shape and appearance quality. Nat. Commun. 2018, 9, 1240, doi:10.1038/s41467-018-03616-y.
  7. Miura, K.; Ikeda, M.; Matsubara, A.; Song, X.J.; Ito, M.; Asano, K.; Matsuoka, M.; Kitano, H.; Ashikari, M. OsSPL14 promotes panicle branching and higher grain productivity in rice. Nat. Genet. 2010, 42, 545-549, doi:10.1038/ng.592.
  8. Che, R.; Tong, H.; Shi, B.; Liu, Y.; Fang, S.; Liu, D.; Xiao, Y.; Hu, B.; Liu, L.; Wang, H.; et al. Control of grain size and rice yield by GL2-mediated brassinosteroid responses. Nat. Plants 2015, 2, 15195, doi:10.1038/nplants.2015.195.
  9. Si, L.; Chen, J.; Huang, X.; Gong, H.; Luo, J.; Hou, Q.; Zhou, T.; Lu, T.; Zhu, J.; Shangguan, Y.; et al. OsSPL13 controls grain size in cultivated rice. Nat. Genet. 2016, 48, 447-456, doi:10.1038/ng.3518.
  10. Ruijter, J.M.; Ramakers, C.; Hoogaars, W.M.; Karlen, Y.; Bakker, O.; van den Hoff, M.J.; Moorman, A.F. Amplification efficiency: linking baseline and bias in the analysis of quantitative PCR data. Nucleic Acids Res. 2009, 37, e45, doi:10.1093/nar/gkp045.
